# #Caremongering: A community-led social movement to address health and social needs during COVID-19

**Hsien Seow**[1]*, **Kayla McMillan**[1], **Margaret Civak**[1], **Daryl Bainbridge**[1], **Alison van der Wal**[2], **Christa Haanstra**[3‡], **Jodeme Goldhar**[4‡], **Samantha Winemaker**[5‡]

**1** Department of Oncology, McMaster University, Hamilton, Ontario, Canada, **2** Department of Health Research Methods, Evidence and Impact, McMaster University, Hamilton, Ontario, Canada, **3** The Change Foundation, Toronto, Ontario, Canada, **4** Dalla Lana School of Public Health, Toronto, Ontario, Canada, **5** Department of Family Medicine, McMaster University, Hamilton, Ontario, Canada

☉ These authors contributed equally to this work.
‡ These authors also contributed equally to this work.
\* seowh@mcmaster.ca

**Data Availability Statement:** All relevant data are within the paper and its Supporting Information files.

## Abstract

### Background

To combat social distancing and stay-at-home restrictions due to COVID-19, Canadian communities began a Facebook social media movement, #Caremongering, to support vulnerable individuals in their communities. Little research has examined the spread and use of #Caremongering to address community health and social needs.

### Objectives

We examined the rate at which #Caremongering grew across Canada, the main ways the groups were used, and differences in use by membership size and activity.

### Methods

We searched Facebook Groups using the term "Caremongering" combined with the names of the largest population centres in every province and territory in Canada. We extracted available Facebook analytics on all the groups found, restricted to public groups that operated in English. We further conducted a content analysis of themes from postings in 30 groups using purposive sampling. Posted content was qualitatively analyzed to determine consistent themes across the groups and between those with smaller and larger member numbers.

### Results

The search of Facebook groups across 185 cities yielded 130 unique groups, including groups from all 13 provinces and territories in Canada. Total membership across all groups as of May 4, 2020 was 194,879. The vast majority were formed within days of the global pandemic announcement, two months prior. There were four major themes identified: personal protective equipment, offer, need, and information. Few differences were found between how large and small groups were being used.

**Funding:** This work was funded by the Labarge Centre for Mobility in Aging COVID-19 Grant. HS received this funding. https://mira.mcmaster.ca/research/research-centre/centre-for-mobility The funders had no role in study design, data collection and analysis, decision to publish, or preparation of the manuscript.

**Competing interests:** The authors have declared that no competing interests exist.

## Conclusions

The #Caremongering Facebook groups spread across the entire nation in a matter of days, engaging hundreds of thousands of Canadians. Social media appears to be a useful tool for spreading community-led solutions to address health and social needs.

## Introduction

The World Health Organization's Ottawa Charter for Health Promotion identifies five strategies for improving health [1, 2]. One of those strategies is strengthening community action, which focuses on empowering communities to set priorities, make decisions, and implement plans to improve health and well-being. This strategy is consistent with other health and social science research such as community development and compassionate communities [3–5]. Community-led solutions have long been an important means to address interconnected health and social issues, such as homelessness and food insecurity [6, 7]. In recent years, social media has become a powerful tool to advance health promotion and communication. Social media can communicate information in a way that spreads quickly and does not require government bureaucracy or financial resources, making it useful for supporting community-led public health approaches [8–10]. However, systematic reviews have concluded that more research is needed to understand social media's reach, efficiency, and impact on the health of a population [11–13].

The COVID-19 global pandemic presents an unprecedented threat to population and individual health. Governments have implemented policies to stop the spread of infection, such as border closures and travel restrictions [14, 15]. Hospitals responded to the pandemic by implementing wide use of protective equipment (PPE), heightened safety procedures, and securing key materials (e.g. ventilators) [16]. In the community however, patient health has been greatly disrupted by self-isolation, social distancing and stay-at-home restrictions. These widespread measures, while necessary to flatten the curve of infection and spread, have exacerbated existing health and social needs in the community, particularly those living with a chronic illness and their families [16–19]. These needs include challenges with social isolation, depressed mood and anxiety, access to primary care and community-based services, financial and food insecurity, and support with activities of daily living (e.g. getting groceries, managing medications, etc.). The pandemic also created new issues for individuals in caring for those with COVID-19 and taking precautions to avoid catching the virus.

COVID-19 presents a unique opportunity to study the role of social media on community-led health initiatives. In Canada, communities began a social media movement, #Caremongering. Within days of the World Health Organization's declaration of the global pandemic on March 11, 2020, the first #Caremongering Facebook Group started in Eastern Canada and inspired communities across Canada to form their own groups [20, 21]. A Facebook Group is a page created by a Facebook member on a topic, intended for other members to join and share this common interest. Members of the group can post content (text, videos, images, etc.) or comment/respond/add to content posted by others [22]. Local #Caremongering Facebook groups formed to help provide vulnerable individuals in their communities with access to food, services, information, and other necessities. Member volunteers deliver supplies and food, donate goods, run errands, or do chores for others, all while maintaining social distancing. The campaign name was inspired by transforming the negative term "scaremongering," to a positive one of "caremongering." This particular social movement might serve as a useful

example of community-led solutions for health care, but its spread and use has not been investigated. This study aims to: 1) examine how far the movement spread and how many Canadians participated in #Caremongering across Canada; 2) characterize how communities used the Facebook groups; and 3) examine differences between big and small-sized community Facebook groups.

## Methods

### Search strategy

We conducted a comprehensive search and examination of #Caremongering Facebook groups in Canada. During the week of May 4th we searched Facebook using the social networking service's built-in search engine. We searched Facebook Groups using the term "caremongering" combined with the name of one of the population centres in Canada, as defined by Statistics Canada [23]. We included the top 50 (by population) population centres in ON, the top 25 in BC, and 10 each in the remaining provinces and territories (total of 185 Canadian cities). This approach ensured that the largest communities, and those most likely to have caremongering groups, across the country were included. We restricted our search to public groups that operated in English.

### Data extraction and analysis

Two analysts (KM and AV) divided the selected population centres and extracted available Facebook analytics on all the groups found: creation date, number of members, average posts per day, change in members past 30 days, change in number of posts past 30 days. We took a purposive sample of the caremongering groups identified for content analysis, as follows: 1) 15 groups with the largest membership, 2) 15 groups with the highest relative posting activity, calculated as the rate of average daily posts per 100 members. The latter tended to be smaller groups ($<$ 600 members) in more rural areas, providing us with a diversity of groups to analyse.

Three analysts (KM, AV, and DB) conducted a content analysis of the 30 groups [24]. Content analysis is a relevant method for making valid inferences as to the manifest content or meaning of text data, including for understanding the contextual use of electronic media. Each analyst read through the group content to determine 1) the description and purpose of the group, 2) group rules, regulations and organization, and, 3) the nature or concept of the content and responses posted.

The analysts took notes and screenshots from each Facebook group examined. The approach to content analysis and classification of posts was based on prior social media research [25, 26]. Each analyst read the posts over the past month of 10 groups (5 large membership and 5 high activity groups each). Specifically, we used a conceptual analysis approach, where the analyst reviewed the text of each post and applied a code through a process of selective reduction summarizing the post as a word or phrase [27]. The analysts proceeded through consecutive posts (100 minimum per group) until they identified no new codes. A constant comparison technique was employed to generate key themes inductively from the codes, first within each Facebook group, and then across all 30 Facebook groups [28]. Throughout data collection the analysts met regularly to compare and discuss their content codes. Once all the selected groups had been examined, the analysts deliberated on consistent themes across the groups and differences between large and high activity (smaller) groups. These preliminary themes were discussed among the authors and finalized. We collected follow-up analytics the week of June 9th from the identified groups. Ethical approval for this study was obtained from the Hamilton Integrated Research Ethics Board (11129). The need for consent to analyze

Facebook posts from members of the groups was waived given that the content was publicly available.

## Results

The search of Facebook groups across 185 cities yielded 130 unique groups, including groups from all 13 provinces and territories across Canada. In the group descriptions, they uniformly described themselves as grassroots networks to assist vulnerable individuals in their communities by offering and seeking support, sharing reliable information, and spreading goodwill in the local community during the COVID-19 pandemic. Many of the groups encouraged members to use prescribed hashtags to indicate the nature of their post, e.g., #ISO (in search of). Group rules varied slightly–though a universally stipulated rule was zero tolerance for posts exhibiting bigotry, scaremongering, or spreading false information. Local residence was not a requirement in most groups given that members may live elsewhere but have joined to help friends or family in that group's geography. Total membership across all 130 groups as of May 4th 2020 was 194,879. The vast majority (96%) were formed within days of the global pandemic announcement. **Fig 1** shows the rapid growth in total members among the 130 Canadian groups from date of group creation (week of March 12 for most groups) to a month later (week of April 12). Membership stablized from this data collection time to subsequent (weeks of May 4 and June 9) and final periods (September 3). As of May, a third (34%) of the groups had over 1000 members, while 14% had less than 100 members. Twenty percent of the groups had at least 20 posts per day on average over the month.

### Analysis of selected groups

**Table 1** shows data from the 30 selected groups. In our sub-analysis, the largest group (Toronto, ON) had 24,822 members; the group had an average of 310 posts per day (average of

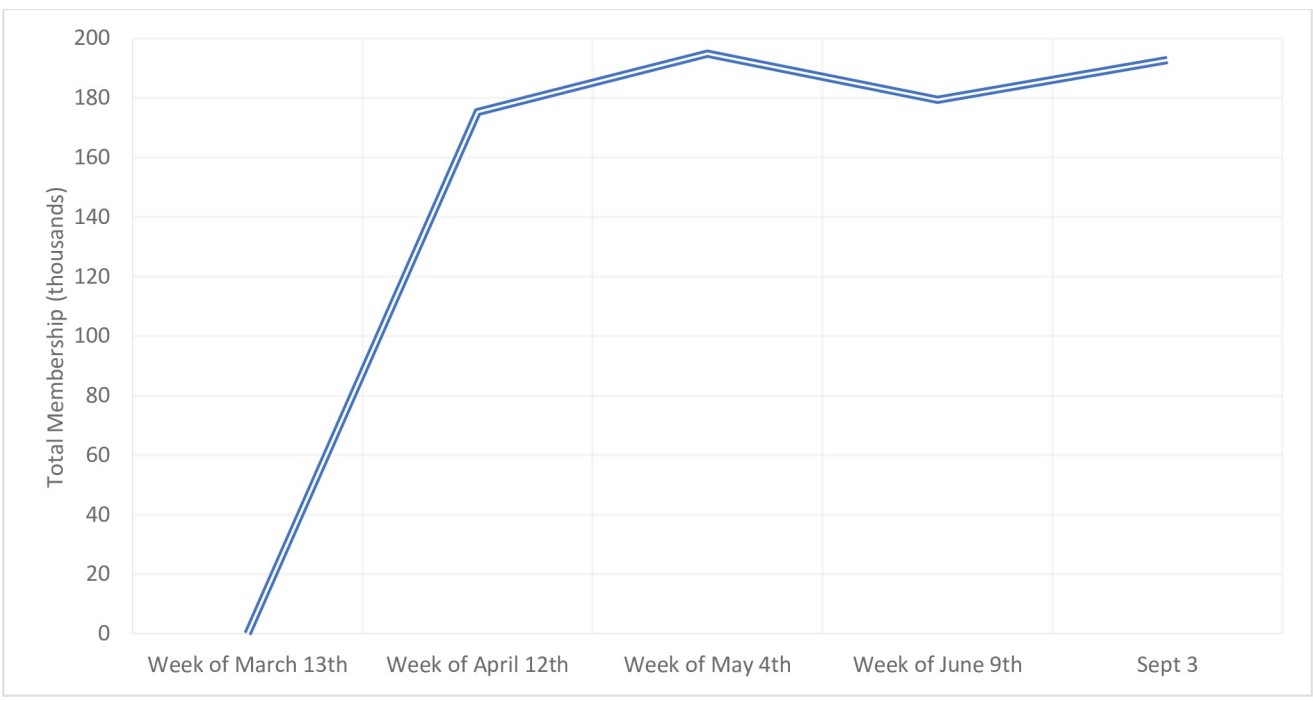

**Fig 1. Facebook membership for 130 Caremongering Facebook groups over time.**

**Table 1. Descriptive information regarding membership and activity of 30 selected #Caremongering Facebook groups, May 2020.**

| City, Province | Population Size | # of Members | Average Posts per Day | Activity/100 members[†] |
|---|---|---|---|---|
| Toronto, ON | 5,429,524 | 24,822 | 310 | 1.25 |
| Kitchener, ON | 470,015 | 8,745 | 80 | 0.91 |
| Hamilton, ON | 693,645 | 6,997 | 80 | 1.14 |
| Ottawa/Gatineau, ON | 989,567 | 6,840 | 40 | 0.58 |
| Saskatoon A, SK | 245,181 | 6,810 | 130 | 1.91 |
| Burlington, ON | 183,315 | 6,287 | 140 | 2.23 |
| Annapolis Valley, NS* | 12,088 | 5,998 | 210 | 3.50 |
| Kingston, ON | 117,660 | 5,232 | 60 | 1.15 |
| Guelph, ON* | 132,397 | 5,229 | 280 | 5.35 |
| Saskatoon B, SK | 245,181 | 4,671 | 50 | 1.07 |
| Niagara Region, ON | 229,246 | 4,304 | 6 | 0.14 |
| Kamloops, BC | 78,026 | 4,020 | 120 | 0.25 |
| Swift Current A, SK | 16,022 | 3,858 | 50 | 1.30 |
| Charlottetown, PEI | 44,739 | 3,642 | 9 | 0.25 |
| London, ON* | 383,437 | 3,534 | 110 | 3.11 |
| Orangeville, ON* | 30,734 | 3,501 | 110 | 3.14 |
| Windsor, ON | 287,069 | 3,213 | 40 | 1.24 |
| Cape Breton, NS | 17,556 | 3,208 | 20 | 0.62 |
| Fredericton, NB | 59,405 | 3,183 | 30 | 0.94 |
| Amherstburg, ON* | 13,910 | 2,622 | 170 | 6.48 |
| Oshawa, ON* | 308,875 | 1,774 | 50 | 2.82 |
| Stratford, ON* | 31,053 | 581 | 30 | 5.16 |
| St. Thomas, ON | 41,813 | 421 | 10 | 2.38 |
| Collingwood, ON | 20,102 | 409 | 10 | 2.44 |
| Midland, ON | 24,353 | 359 | 10 | 2.79 |
| Swift Current B, SK | 16,022 | 357 | 10 | 2.80 |
| Montreal, QB* | 1,704,694 | 272 | 10 | 3.68 |
| Peggy's Cove, NS* | 30 | 223 | 9 | 4.04 |
| York Region, ON* | 1,109,909 | 183 | 7 | 3.83 |
| Woodstock, ON | 40,404 | 116 | 3 | 2.59 |

*denotes Facebook groups selected based on highest activity

[†]Activity rate = average daily posts/number of members x 100).

"Small" facebook groups in our sample were considered to be those with less than 600 members.

1.25 posts/100 members). The smallest group (Woodstock, ON) had 116 members; the group had an average of 3 posts per day (average of 2.59 posts/100 members). The most active group (Amherstburg, ON) had 2,622 members; that group had an average of 170 posts per day (average of 6.48 posts/100 members).

## How members used the Caremongering Facebook groups

The research team identified 4 major themes within the posts of the 30 groups (**Table 2**). The themes coincided along the broad categories of hashtags (#) used, regardless of whether the use of hashtags was an explicit rule in the Facebook group. A hashtag is common social media tool used to organize posts and content. The most common hashtags found by the research team were: #offer, #iso (in search of), #discussion and #info.

**Table 2. Main themes and hashtag use in Caremongering Facebook groups.**

| Theme | Category | Associated Hashtags |
|---|---|---|
| 1.Personal protective equipment (PPE) | e.g. PPE offers, instructions on making masks, sale and delivery of masks, hand sanitizer, and disinfectant wipes. | #resources |
| | | #offer |
| 2. Offer | Offering Materials | #offer |
| | e.g. food, clothes, and recreational activities. | #donate |
| | Offering Services | #offer |
| | e.g. pick-up of essential materials, such as prescriptions, drop off of groceries, etc. | #communitycare |
| 3. Need | Need for materials | #iso |
| | e.g. food, pantry supplies, clothes, air conditioners, etc. | #help |
| | Need for Services | #need |
| | e.g. assistance with pets, taxes, delivery, transportation to medical appointments, etc. | |
| 4. Information | Community Information | #shops |
| | e.g. news updates, store closures, restaurant take-out hours, etc. | #thingstodo |
| | COVID General Info | #municipalchanges |
| | e.g. cases, test centers, outbreaks, etc. | #news |
| | Positive/Inspirational | #mentalhealth |
| | e.g. good news stories, acts of kindness, etc. | #covidnews |
| | Discussion/ Advice | #municipalchanges |
| | e.g. employment insurance payments, activities and schedules for home schooling children, etc. | #smile |
| | | #goodbusiness |
| | | #sharethelove |
| | | #thankyou |
| | | #discussion |
| | | #resources |

**Theme 1: PPE.** Personal Protective Equipment (PPE) was a recurring theme among all the Facebook groups analyzed. PPE includes masks, (both reusable and disposable) and equipment and supplies for antibacterial cleaning and disinfection. Some Facebook groups made making homemade masks a central purpose, which then expanded to delivering masks to Long Term Care homes in desperate need of equipment for healthcare providers. This theme included selling or donating PPE, the organization of members sewing masks as a collective, and offering supplies for and information on mask making, including sharing recommendations from the Centres for Disease Control on fabric selection, adding filters, and the use of "ear savers". Example: *"With Trudeau suggesting masks today I have been busy sewing more masks for those who need them."*

**Theme 2: Offer.** The second major theme was "offer." This encompassed individuals or groups offering something for those in need. This major theme had multiple manifestations across the 30 groups, however the research team identified two main categories: offers of material resources or offering of services. Offering of material resources could include offering food, pantry supplies (e.g. for baking), clothing, cooked food, recreational activity (including puzzles or activities for children who were not in school). Example: *"I made some more vegan stuffing and have extra—direct message me."* Offering of services ranged from offering of picking up prescriptions and groceries, to offering assistance with resume building for those who had lost their jobs. Example: "*Student can drive those requiring assistance to groceries, appointments, etc.*" Many groups posted guidelines for offering grocery delivery or prescription pick up to those who were interested in doing so, in terms of money exchanging hands and keeping the safety of others in mind.

**Theme 3: Need.**   The third major theme identified was individuals or groups posting about needs. If the group used hashtags, this was often denoted as #iso (in search of), #need or #help. These posts also fell into two main categories: need for services or need for material goods. Some of the services requested included grocery delivery or pick up or assisting the elderly with tax preparation. Example: "*#ISO tax assistance for two elderly people.*" Some of the goods people needed included food and groceries, and household cleaning supplies. Example: "*#ISO Is anyone able to donate some basic items (frozen fruit etc.)*? *We are still struggling through this pandemic.*"

**Theme 4: Information.**   The fourth major theme was that of requested or provided information. This theme contained a wide variety of content, which the research team compiled into four categories:

1. Community Information: This included information about local park closures, information about businesses that are still open and local services that are helping people. Example: "*Does anyone know any groomers that are still taking dogs*?"

2. COVID Information: This included information about COVID-19 cases in an area, hospital information and guidelines about getting a COVID-19 test. Example: "*N.B. COVID-19 roundup*: *Province expands testing protocols.*"

3. Positive/ Inspiration: This category included positive messaging for front line workers, memes or jokes or acknowledging the charitable efforts of individuals, organizations, and businesses contributing to local communities. Example: "*Public health called today to give me clearance. This past few weeks has taught me so many lessons.*"

4. Discussion/ Advice: This category included advice about the Canada Emergency Response Benefit (CERB), Employment Insurance related questions and general advice about navigating COVID-19. Example: "*How many days for EI (employment insurance) payments to come through*?"

## Small vs large groups

Our analysis of posts in the 30 groups revealed there were not major differences in themes between how the 21 large groups and 9 small groups were being used. Regardless of group membership size, they were able to create a feeling of a smaller tightly-knit community within larger geographic areas. For instance, smaller groups posted about delivering home-cooked foods to other. Similar offers and connections were present in large groups, although posting members would stipulate neighborhoods or intersections, for example, "X available, located at King and Main St." Large groups tended to have more posts per day, thus had more variety of content. The themes we identified were consistent in groups across provinces.

## Discussion

Our study of the #Caremongering Facebook groups found that the volunteer social media movement spread to at least 130 communities—both big and small—engaging over 190,000 Canadians within days of the COVID-19 emergency declaration. These groups spread to every province and territory across Canada. Caremongering groups share a common purpose in providing an online platform for sharing of resources to individuals in their communities and the posting of local and national information related to the pandemic. The size of both the groups and the communities varied greatly, yet the themes were consistently around PPE, offers for things, request for things, and information sharing. The exponential and rapid growth of these networks demonstrates the reach and efficiency of using social media to

develop and implement community-led solutions virtually, which was especially beneficial considering the widespread stay-at-home and social distancing orders of the pandemic.

Our results showed that #Caremongering groups provided direct health information (e.g. announcements about public health safety and testing sites), as well as addressed some broader social needs (e.g., unemployment benefits, food bank donations, etc.). There are some parallels between the #Caremongering social media intervention and other effective health promotion strategies, for instance the Australian response to HIV/AIDS epidemic in the 1980s. Specifically, grassroots and informal community mobilization and advocacy were critical to improving care, but also enhancing access to social and health prevention resources, which were largely responsible for the decline in HIV incidence in Australia [29–31]. Unfortunately, more recent systematic reviews on the specific role of social media on health issues are generally narrow in their research focus and outcomes, such as the ability of social media to promote HIV testing and medication adherence or recruit subjects to participate in smoking cessation programs [32, 33]. Thus, existing evidence is generally limited in addressing the multifaceted impacts of social media, such as social needs, community mobilization, or government response.

To explore whether #Caremongering is a useful health promotion tool to strengthen community action, we refer back to the World Health Organization criteria: [2] it states effective health promotion strategies must fulfill three basic prerequisites: Advocate, Enable and Mediate [34]. "Advocate" is the ability to promote favorable conditions through advocacy for health. "Enable" relates to ensuring equal opportunities and resources to allow everyone to achieve their fullest health potential. "Mediate" is the coordination of action to promote health by multiple sectors, not only health. #Caremongering allowed participants to advocate for and request supports and services they needed easily, often by using #need or #iso on the group. #Caremongering was also created to enable health equity and support vulnerable individuals in local communities. Finally, it served to mediate multiple sectors, diverse individuals and businesses in a geographic community to work together toward a common goal. While the effectiveness of #Caremongering itself as a health promotion tool requires more research beyond this initial study, it appears social media has promising potential to greatly support health promotion, including implementing activities that address physical, mental, and social well-being.

Our study has other limitations. Our search was limited to larger population areas and to English language or bilingual groups, and therefore, our findings may not be representative of remote, new immigrant, or French language groups. We examined changes in group membership over time but not the level of activity or content of posting, which may have varied since data collection. Almost all 130 groups examined were still active by the September period, however members may have retained their membership even if they were not longer active in the group. Our content analysis represents a snapshot of 30 groups that may not fully represent the overall content posted among all the caremongering groups we identified; however, we believe our sampling approach feasibly captured group diversity. Further research is required to understand the impact of the support provided on individual's quality-of-life and other health outcomes, as well as how these groups help specific vulnerable populations (e.g. older adults, homeless, etc.). We also did not assess the sustainably of this movement, which rests upon the continued activity of site moderators and members.

In conclusion, our study showed that the #Caremongering social media movement quickly mobilized and engaged tens of thousands of people within a few days to offer support to others, even within small communities. Convening for a shared purpose over social media is a powerful means by which communities can address complex problems that cannot be resolved without shared responsibility with individual citizens and joint action. Social movements, fueled by social media, can be an important public health tool to support the health of vulnerable populations in the community.

## Supporting information

**S1 Table. Caremongering group data.**
(DOCX)

## Author Contributions

**Conceptualization:** Hsien Seow, Kayla McMillan, Margaret Civak, Daryl Bainbridge.

**Data curation:** Kayla McMillan, Daryl Bainbridge, Alison van der Wal.

**Formal analysis:** Hsien Seow, Kayla McMillan, Daryl Bainbridge, Alison van der Wal.

**Funding acquisition:** Hsien Seow, Kayla McMillan, Margaret Civak, Daryl Bainbridge.

**Investigation:** Hsien Seow, Kayla McMillan, Daryl Bainbridge.

**Methodology:** Hsien Seow, Kayla McMillan, Daryl Bainbridge.

**Project administration:** Hsien Seow, Daryl Bainbridge.

**Resources:** Hsien Seow.

**Supervision:** Hsien Seow.

**Validation:** Hsien Seow.

**Writing – original draft:** Hsien Seow, Kayla McMillan, Daryl Bainbridge, Alison van der Wal.

**Writing – review & editing:** Hsien Seow, Kayla McMillan, Margaret Civak, Daryl Bainbridge, Alison van der Wal, Christa Haanstra, Jodeme Goldhar, Samantha Winemaker.

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
