## [Decision Letter · Decision Letter 0]

30 Oct 2020

PONE-D-20-28663

#Caremongering: A community-led social movement to address health and social needs during COVID-19

PLOS ONE

Dear Dr. Seow,

Thank you for submitting your manuscript to PLOS ONE. After careful consideration, we feel that it has merit but does not fully meet PLOS ONE’s publication criteria as it currently stands. Therefore, we invite you to submit a revised version of the manuscript that addresses the points raised during the review process.

We look forward to receiving your revised manuscript.

Kind regards,

Holly Seale

Academic Editor

PLOS ONE

Journal Requirements:

2. We note that you have indicated that consent was not obtained as no participants were involved in data collection. Please indicate whether your ethics committee waived the requirement to obtain consent from participants involved in the study (i.e. those who you analysed the posts of).

Reviewers' comments:

Reviewer's Responses to Questions

**Comments to the Author**

1. Is the manuscript technically sound, and do the data support the conclusions?

Reviewer #1: Partly

Reviewer #2: Yes

2. Has the statistical analysis been performed appropriately and rigorously? 

Reviewer #1: I Don't Know

Reviewer #2: Yes

3. Have the authors made all data underlying the findings in their manuscript fully available?

Reviewer #1: No

Reviewer #2: Yes

4. Is the manuscript presented in an intelligible fashion and written in standard English?

Reviewer #1: Yes

Reviewer #2: Yes

5. Review Comments to the Author

Reviewer #1: Thank you for the opportunity to review this paper. It was clear and well written and does a good job of documenting how people interacted with a social movement at a certain point in time of the COVID-19 pandemic.

The only part of the paper that I was not convinced by was the methods. The search strategy and data collection section was clear but the method of the analysis was not. The author writes that they based their method on previous work but don’t explain what the methodology is and why it is appropriate for this work. I expected to see a citation of the methodology they followed here but was referred to (I assume) the author’s previous studies. I don’t think the reader expects to have to read other work from the author to understand how this study was done - it would be great to have the methods adequately explained in this section so that the paper can be assessed on its own merits.

Also on methods, the author reports that they analysed content in order to “determine 1) the description and purpose of the group, 2) group rules, regulations and organization, and, 3) the nature of the content posted, as well as responses to posts.” They only report on the third aspect of this. It could be that the first point was to determine eligibility (useful to know if there were things other than language that would rule a group in/out). The second is interesting – did they all have the same rules? Were there different rules for paid and unpaid assistance etc? What sorts of things would make a person ineligible for the group? Did they have to live locally etc? This may be beyond scope! But I wondered all of these things as I was reading, because I had anticipated they would be reported as signposted in the second to last paragraph of the methods section.

Finally, on methods, it was not clear to me what the following sentence meant, particularly the first part: “The topic of each post reviewed was coded, proceeding through subsequent posts (100 minimum per group) until no new codes emerged.” Can you rewrite this so that is it clearer?

Other things I wondered and expected to find as I read the paper – did activity levels stay the same over time? Membership was reported at different points between March and Sept. What time period did the ‘posts per day’ reflect? Membership and activity aren’t the same – I would imagine activity would have waned over time as people became more accustomed to living in a pandemic situation, and this seems relevant to your study given that you gathered data at different points. Membership might have remained stable because people often don't bother to leave groups they are not actively engaged with. Can you report on this?

While I completely understand not having the capacity to analyse groups conducted in languages other than English, it is probably a limitation also in that it may have missed groups of newer migrants, who might have had different needs from other communities. This could be reported as a limitation.

Finally, I wasn’t convinced by the argument in the conclusion that the caremongering groups are health promotion tools. They might be but the claim seemed to come from nowhere and did not add to the paper. The paper seems a useful description of community rallying around a certain cause at a certain point in time. To make it more needs further analysis and argument I think.

Note that while I said all the data aren’t available (per the compulsory question) I do not consider that to be an issue. It can be a problem, reviewing qualitative work in systems set up for quantitative assessment. The data are not required I don’t believe, and it is difficult to imagine how they could ethically be retained given that they are identifiers. So this is more a note for the editor.

Reviewer #2: This study provides a descriptive account of Facebook-hosted online community support hubs that emerged across Canada in response to COVID-19 lockdowns. The authors describe the phenomenon and discuss whether the activities these hubs foster meet the definition of ‘health promotion’, concluding that they do.

Study methods are well described, and I found questions that had about these (impacts of English only; underrepresentation of remote populations; and the ‘snapshot’ approach) were appropriately addressed in the limitations section.

I note that there are no page or line numbers on the manuscript, which does make it harder to identify where revisions are recommended.

I have a number of recommendations to make, which I will list below.

1. Please include a reference for ‘content analysis’ so that readers can see how your approach to data analysis is informed. Providing a brief description of why you chose this approach would also be welcome.

2. Avoid using the word ‘emerged’ regarding themes, as this term underestimates the role of the researcher in the identification of themes. ‘We identified’ is a good substitute.

3. Typo: ‘codese’

4. In the results, the phrases ‘members who posted 310 times per day’ and ‘170 time per day’ should be rewritten to make ti clear that these were the total number of posts on a day, not the average contribution of each member of the hub.

5. Figure 1 should be ‘table 1’

6. Discussion: the sentence beginning ‘whereas’ needs revision (it currently reads as part of the previous sentence, into which a full stop has been interpolated to make the sentences shorter).

7. Currently, the authors reference social media interventions on smoking cessation, breast cancer and HIV, and make the comment that #Caremongering is not focused on a siloed health issue, but on the ‘existing health and social needs in the community during an unprecedented global pandemic’. The reader is left to assume that these other social media interventions are based narrowly on health issues, but the reference for these are systematic reviews of – narrowly defined health issues! I think there is a real opportunity missed here to make some comparisons between #Caremongering and the community responses to HIV, which is far richer than a systematic review on specific topics on social media can capture. Consider some of these sources: Brown, G., et al., Mobilisation, politics, investment and constant adaptation: lessons from the Australian health-promotion response to HIV. Health Promotion Journal of Australia, 2014. 25(1): p. 35-41. https://theinstituteofmany.org/ Plummer D, Irwin L (2006) Grassroots activities, national initiatives, and HIV prevention: clues to explain Australia’s dramatic early success in controlling the HIV epidemic. Int J STD AIDS 17, 787–93. Grant, M.P., et al., Communal responsibility: a history of health collectives in Australia. Internal Medicine Journal, 2019. 49(9): p. 1177-1180. If this is too complex a task , the authors should revise their comments specifically about the narrow focus of HIV social media to reflect how the way that research counts social media (ie by focusing narrowly) may obfuscate anything that occurs outside a narrow lens.

6. PLOS authors have the option to publish the peer review history of their article (what does this mean?). If published, this will include your full peer review and any attached files.

Reviewer #1: No

Reviewer #2: **Yes: **Bridget Haire

---

## [Author Response · Author response to Decision Letter 0]

1 Dec 2020

We wish to thank the editors and reviewers for their careful consideration of this manuscript. We agreed with the reviewers’ remarks and have attempted to address all their questions and suggested revisions item-by-item as follows. 

REPLY: This has been done. Correct PLOS One formatting and referencing applied.

p2. We note that you have indicated that consent was not obtained as no participants were involved in data collection. Please indicate whether your ethics committee waived the requirement to obtain consent from participants involved in the study (i.e. those who you analysed the posts of).

REPLY: We have written in the methods an explicit statement saying: “The need for consent to analyze Facebook posts from members of the groups was waived given that the content was publicly available.”

p3. Your ethics statement should only appear in the Methods section of your manuscript. If your ethics statement is written in any section besides the Methods, please delete it from any other section.

REPLY: This has been done. 

Reviewer's Responses to Questions

Reviewer #1: 

Thank you for the opportunity to review this paper. It was clear and well written and does a good job of documenting how people interacted with a social movement at a certain point in time of the COVID-19 pandemic.

The only part of the paper that I was not convinced by was the methods. The search strategy and data collection section was clear but the method of the analysis was not. The author writes that they based their method on previous work but don’t explain what the methodology is and why it is appropriate for this work. I expected to see a citation of the methodology they followed here but was referred to (I assume) the author’s previous studies. I don’t think the reader expects to have to read other work from the author to understand how this study was done - it would be great to have the methods adequately explained in this section so that the paper can be assessed on its own merits.

REPLY: Thank you for this comment. Consistent with the other reviewer’s comments, we have elaborated further on our qualitative methods in the following ways:

-We have added a citation for “content analysis” (Krippendorff, 2018)

-We have added a statement of why we chose this approach: “Content analysis is a relevant method for making valid inferences as to the manifest content or meaning of text data, including for understanding the contextual use of electronic media.” 

-We have also specified the type of content analysis we used, i.e. conceptual analysis, and included a reference: “Specifically, we used a conceptual analysis approach, where the analyst reviewed the text of each post and applied a code through a process of selective reduction summarizing the post as a word or phrase” (Wilson, 2016). 

References:

• Krippendorff K. Content analysis: An introduction to its methodology: Sage publications; 2018.

• Wilson V. Research methods: Content analysis. Evidence Based Library and Information Practice 2016, 11(1 (S)):41-43.

Also on methods, the author reports that they analysed content in order to “determine 1) the description and purpose of the group, 2) group rules, regulations and organization, and, 3) the nature of the content posted, as well as responses to posts.” They only report on the third aspect of this. It could be that the first point was to determine eligibility (useful to know if there were things other than language that would rule a group in/out). The second is interesting – did they all have the same rules? Were there different rules for paid and unpaid assistance etc? What sorts of things would make a person ineligible for the group? Did they have to live locally etc? This may be beyond scope! But I wondered all of these things as I was reading, because I had anticipated they would be reported as signposted in the second to last paragraph of the methods section.

REPLY: As suggested, we added a statement about the first 2 steps in the Results:

“Many of the groups encouraged members to use prescribed hashtags to indicate the nature of their post, e.g., #ISO (in search of). Group rules varied slightly – though a universally stipulated rule was zero tolerance for posts exhibiting bigotry, scaremongering, or spreading false information. Local residence was not a requirement in most groups, given that members may live elsewhere but have joined to help friends or family in that group’s geography.” Note, none of the 30 sites we selected, where we did content analysis, were excluded.

Finally, on methods, it was not clear to me what the following sentence meant, particularly the first part: “The topic of each post reviewed was coded, proceeding through subsequent posts (100 minimum per group) until no new codes emerged.” Can you rewrite this so that is it clearer?

REPLY: We have revised this sentence and the preceding one. They now read as:

“Specifically, we used a conceptual analysis approach, where the analyst reviewed the text of each post and applied a code through a process of selective reduction summarizing the post as a word or phrase.[27] The analysts proceeded through consecutive posts (100 minimum per group) until they identified no new codes.”

Other things I wondered and expected to find as I read the paper – did activity levels stay the same over time? Membership was reported at different points between March and Sept. What time period did the ‘posts per day’ reflect? Membership and activity aren’t the same – I would imagine activity would have waned over time as people became more accustomed to living in a pandemic situation, and this seems relevant to your study given that you gathered data at different points. Membership might have remained stable because people often don't bother to leave groups they are not actively engaged with. Can you report on this?

REPLY: ‘Posts per day’ reflect Facebook analytics in April 2020, indicated in the Table 1 title. 

Activity may have waned over time even if membership remained stable – we have added this as a study limitation. “Almost all 130 groups examined were still active by the September period, however members may have retained their membership even if they were not longer active in the group." Reporting on activity levels over time (i.e. posts/group over time) is beyond the scope of this study and listed as a limitation: “We examined changes in group membership over time but not the level of activity or content of posting, which may have varied since data collection.”

While I completely understand not having the capacity to analyse groups conducted in languages other than English, it is probably a limitation also in that it may have missed groups of newer migrants, who might have had different needs from other communities. This could be reported as a limitation.

REPLY: The exclusion of non-English groups and new immigrant groups is reported as a limitation.

Finally, I wasn’t convinced by the argument in the conclusion that the caremongering groups are health promotion tools. They might be but the claim seemed to come from nowhere and did not add to the paper. The paper seems a useful description of community rallying around a certain cause at a certain point in time. To make it more needs further analysis and argument I think.

REPLY: We have changed the topic sentence to better link Caremongering to our introduction, which began with the WHO Charter for Health Promotion criteria: “To explore whether #Caremongering is a useful health promotion tool to strengthen community action, we refer back to the World Health Organization criteria: it states effective health promotion strategies must fulfill three basic prerequisites: Advocate, Enable and Mediate” We hope this make the linkage more clear. Second, we agree that our study does not definitively make the case of Caremongering as an effective health promotion tool/strategy. Therefore, we have softened the conclusion of this paragraph to speak of the potential of the broader topic of social media movements. It now reads: “While the effectiveness of #Caremongering itself as a health promotion tool requires more research beyond this initial study, it appears social media has promising potential to greatly support health promotion, including implementing activities that addressed physical, mental, and social well-being.”

Note that while I said all the data aren’t available (per the compulsory question) I do not consider that to be an issue. It can be a problem, reviewing qualitative work in systems set up for quantitative assessment. The data are not required I don’t believe, and it is difficult to imagine how they could ethically be retained given that they are identifiers. So this is more a note for the editor.

REPLY: Thank you for your comment.

Reviewer #2: 

This study provides a descriptive account of Facebook-hosted online community support hubs that emerged across Canada in response to COVID-19 lockdowns. The authors describe the phenomenon and discuss whether the activities these hubs foster meet the definition of ‘health promotion’, concluding that they do.

Study methods are well described, and I found questions that had about these (impacts of English only; underrepresentation of remote populations; and the ‘snapshot’ approach) were appropriately addressed in the limitations section.

I note that there are no page or line numbers on the manuscript, which does make it harder to identify where revisions are recommended.

REPLY: We have added page numbers.

I have a number of recommendations to make, which I will list below.

1. Please include a reference for ‘content analysis’ so that readers can see how your approach to data analysis is informed. Providing a brief description of why you chose this approach would also be welcome.

REPLY: We have added a reference for ‘content analysis’ (Krippendorff , 2018) and added a statement of why we chose this approach: “Content analysis is a relevant method for making valid inferences as to the manifest content or meaning of text data, including for understanding the contextual use of electronic media.” As well, we specify the type of content analysis we used, i.e. conceptual analysis: “Specifically, we used a conceptual analysis approach, where the analyst reviewed the text of each post and applied a code through a process of selective reduction summarizing the post as a word or phrase” (Wilson, 2016). 

References:

• Krippendorff K. Content analysis: An introduction to its methodology: Sage publications; 2018.

• Wilson V. Research methods: Content analysis. Evidence Based Library and Information Practice 2016, 11(1 (S)):41-43.

2. Avoid using the word ‘emerged’ regarding themes, as this term underestimates the role of the researcher in the identification of themes. ‘We identified’ is a good substitute.

REPLY: As suggested, we have changed ‘emerged’ to ‘we identified’.

3. Typo: ‘codese’

REPLY: Typo corrected to ‘codes’.

4. In the results, the phrases ‘members who posted 310 times per day’ and ‘170 time per day’ should be rewritten to make it clear that these were the total number of posts on a day, not the average contribution of each member of the hub.

REPLY: The sentences have been revised as: “…the largest group (Toronto, ON) had 24,822 members; the group had an average of 310 posts per day (average of 1.25 posts/100 members). The smallest group (Woodstock, ON) had 116 members; the group had an average of 3 posts per day (average of 2.59 posts/100 members.”

5. Figure 1 should be ‘table 1’

REPLY: Figure 1 is a line graph and therefore we have labeled it as a figure.

6. Discussion: the sentence beginning ‘whereas’ needs revision (it currently reads as part of the previous sentence, into which a full stop has been interpolated to make the sentences shorter).

REPLY: Sentences revised.

7. Currently, the authors reference social media interventions on smoking cessation, breast cancer and HIV, and make the comment that #Caremongering is not focused on a siloed health issue, but on the ‘existing health and social needs in the community during an unprecedented global pandemic’. The reader is left to assume that these other social media interventions are based narrowly on health issues, but the reference for these are systematic reviews of – narrowly defined health issues! I think there is a real opportunity missed here to make some comparisons between #Caremongering and the community responses to HIV, which is far richer than a systematic review on specific topics on social media can capture. Consider some of these sources: 

• Brown, G., et al., Mobilisation, politics, investment and constant adaptation: lessons from the Australian health-promotion response to HIV. Health Promotion Journal of Australia, 2014. 25(1): p. 35-41. https://theinstituteofmany.org/ Plummer D, Irwin L (2006) 

• Grassroots activities, national initiatives, and HIV prevention: clues to explain Australia’s dramatic early success in controlling the HIV epidemic. Int J STD AIDS 17, 787–93. 

• Grant, M.P., et al., Communal responsibility: a history of health collectives in Australia. Internal Medicine Journal, 2019. 49(9): p. 1177-1180. 

If this is too complex a task , the authors should revise their comments specifically about the narrow focus of HIV social media to reflect how the way that research counts social media (i.e. by focusing narrowly) may obfuscate anything that occurs outside a narrow lens.

REPLY: Thank you for this comment. We have read the suggested sources, and reconsidered the intent of this paragraph. As a result, we have made major revisions to this paragraph to incorporate the above points and references. It now reads:

“Our results showed that #Caremongering groups provided direct health information (e.g. announcements about public health safety and testing sites), as well as addressed some broader social needs (e.g., unemployment benefits, food bank donations, etc.). There are some parallels between the #Caremongering social media intervention and other effective health promotion strategies, for instance the Australian response to HIV/AIDS epidemic in the 1980s. Specifically, grassroots and informal community mobilization and advocacy were critical to improving care, but also enhancing access to social and health prevention resources, which were largely responsible for the decline in HIV incidence in Australia.(CITE) Unfortunately, more recent systematic reviews on the specific role of social media on health issues are generally narrow in their research focus and outcomes, such as the ability of social media to promote HIV testing and medication adherence or recruit subjects to participate in smoking cessation programs. (CITE) Thus, existing evidence is generally limited in addressing the multifaceted impacts of social media, such as social needs, community mobilization, or government response.”

---

## [Decision Letter · Decision Letter 1]

4 Jan 2021

#Caremongering: A community-led social movement to address health and social needs during COVID-19

PONE-D-20-28663R1

Dear Dr. Seow,

We’re pleased to inform you that your manuscript has been judged scientifically suitable for publication and will be formally accepted for publication once it meets all outstanding technical requirements.

Kind regards,

Holly Seale

Academic Editor

PLOS ONE

Additional Editor Comments (optional):

Reviewers' comments:

Reviewer's Responses to Questions

**Comments to the Author**

1. If the authors have adequately addressed your comments raised in a previous round of review and you feel that this manuscript is now acceptable for publication, you may indicate that here to bypass the “Comments to the Author” section, enter your conflict of interest statement in the “Confidential to Editor” section, and submit your "Accept" recommendation.

Reviewer #1: All comments have been addressed

Reviewer #2: All comments have been addressed

2. Is the manuscript technically sound, and do the data support the conclusions?

Reviewer #1: (No Response)

Reviewer #2: Yes

3. Has the statistical analysis been performed appropriately and rigorously? 

Reviewer #1: (No Response)

Reviewer #2: N/A

4. Have the authors made all data underlying the findings in their manuscript fully available?

Reviewer #1: (No Response)

Reviewer #2: Yes

5. Is the manuscript presented in an intelligible fashion and written in standard English?

Reviewer #1: (No Response)

Reviewer #2: Yes

6. Review Comments to the Author

Reviewer #1: (No Response)

Reviewer #2: Thanks for addressing my comments

000000000000000000000000000000000000000000000000000000000000000000

7. PLOS authors have the option to publish the peer review history of their article (what does this mean?). If published, this will include your full peer review and any attached files.

Reviewer #1: **Yes: **Jane Williams

Reviewer #2: **Yes: **Bridget Haire

---

## [Editor Report · Acceptance letter]

7 Jan 2021

PONE-D-20-28663R1 

#Caremongering: A community-led social movement to address health and social needs during COVID-19 

Dear Dr. Seow:

I'm pleased to inform you that your manuscript has been deemed suitable for publication in PLOS ONE. Congratulations! Your manuscript is now with our production department. 

Kind regards, 

on behalf of

Dr. Holly Seale 

Academic Editor

PLOS ONE